# Genome-Wide Identification and Expression Analysis of the MADS-Box Family in *Ginkgo biloba*

**Ke Yang** [1,†], **Zhongbing Liu** [2,†], **Xueyin Chen** [1], **Xian Zhou** [1], **Jiabao Ye** [1,*], **Feng Xu** [1], **Weiwei Zhang** [1], **Yongling Liao** [1], **Xiaoyan Yang** [1] **and Qijian Wang** [1]

1   College of Horticulture and Gardening, Yangtze University, Jingzhou 434025, China
2   School of Horticulture and Landscape, Wuhan University of Bioengineering, Wuhan 430415, China
*   Correspondence: yejiabao@yangtzeu.edu.cn
†   These authors contributed equally to this work.

**Abstract:** As the most significant transformation stage of plants, the flowering process has typically been the focus of research. MADS-box gene plays an important regulatory role in flower development. In this study, 26 MADS-box genes were identified from *Ginkgo biloba*, including 10 type-I genes and 16 type-II genes, which were distributed on eight chromosomes. There was no collinearity between the *GbMADS* genes, and the homology with genes from other species was low. All GbMADS proteins contain conserved MADS domains. The gene structures of *GbMADS* in the same gene family or subfamily differed, but the conserved protein motifs had similar distributions. The microRNA (miRNA) target sites of the *GbMADS* genes were predicted. It was found that the expression of 16 *GbMADS* genes may be regulated by miRNA. The results of *cis*-acting element analysis showed that the 26 *GbMADS* genes contained a large number of hormones regulated and light-responsive elements as well as stress-response elements. Furthermore, the quantitative real-time PCR (qRT-PCR) experimental results showed that most *GbMADS* genes were differentially expressed in the male and female flowers at different developmental stages. Among them, the only MIKC * gene *GbMADS16* has the highest expression in the metaphase development of the microstrobilus (M2) stage and is almost not expressed in female flowers. Taken together, these findings suggest that the MADS-box genes may play an important role in the development and differentiation of *G. biloba* flowers.

**Keywords:** MADS-box gene; *Ginkgo biloba*; phylogenetic analysis; miRNA target site; flower development; expression analysis



## 1. Introduction

*Ginkgo biloba* L. is a gymnosperm, the only extant member of Ginkgoaceae, known as the "living fossil" of the plant kingdom [1]. It has high medicinal value and is one of the world's most widely used natural medicines, with high expectations to contribute significantly to health care and medicine in the future. The leaves are rich in flavonoids, terpene trilactones, and other active ingredients [2,3], and their extracts have significant curative effects on cardiovascular diseases, hypertension, and atherosclerosis [4–6]. The fruit can be used as a medicine or can be consumed which has the ability to clear the lungs, expectorant, improve brain function, and also have antibacterial properties, and is used to treat cough, asthma, enuresis, and Alzheimer's disease [7,8]. However, the long juvenile phase hindered the development of suitable varieties. Although breeders have attempted to shorten the juvenile phase using cutting and grafting, the plants typically take 5–10 years to bear fruit, which still constitutes a long breeding cycle for trees. Therefore, new methods are needed to further shorten its juvenile phase.

MADS-box is a homeotic gene family that formed two lineages, namely type-I and type-II during evolution [9]. In plants, type-I was further divided into Mα, Mβ, and Mγ, while type-II (MIKC) was divided into MIKC * and MIKCᶜ [10,11]. Both types contain a

MADS domain, while type-II also contains an intervening (I) region, keratin (K) box, and C-terminal [12]. These domains promote the binding of dimers and cofactors, form complexes between codimers and DNA, and mediate the interaction between proteins [11,13,14]. In addition, the C-terminal can activate transcriptional processes, and differences in the C-domain lead to different functions of MADS-box protein [15]. Based on these different functions, MADS-box genes in angiosperms can be divided into *AP1/SQUA* (Class A), *AP3/PI* and *GGM13* (Class B), *AG* (Class C), *STK/AGL11* (Class D), *AGL6/SEP* and *AGL2* (Class E), and S and P branches [16–19]. Melzer et al., further classified the MADS-box genes of plants into 14 main evolutionary branches: *AGAMOUS* (*AG*), *AGL2* (*SEP*), *AGL6*, *AGL12*, *AGL15*, *AGL17*, *AP3/PI* (*DEF/GLO*), *GGM13*, *STMADS11* (*SVP*), *AP1* (*SQUA*), *FLC*, *TM3*, *TM8*, and *GpMADS4* [20]. Currently, research on the MADS-box genes of gymnosperms has mainly concentrated on conifers and Gnetophyta, and the obtained genes are mainly divided into the *DEF/GLO*, *GGM13*, *AG*, *AGL6*, *TM3*, and *TM8* subgroups [21–24]. A few genes on the *AG*, *SEP*, and *GGM13* branches have been reported in *G. biloba* [25].

In recent years, molecular biology research on the developmental mechanisms of higher plants has been gradually carried out. As the most significant transformation stage of plants, the flowering process has typically been the focus of research. MADS-box genes are key integration genes in the plant flowering regulation network that have been reported in many plants, such as *Arabidopsis thaliana* [26], *Antirrhinum majus* [27], *Populus trichocarpa* [28], *Brassica rapa* [29], *Ananas comosus* [30], and *Malus domestica* [31]. The MADS-box genes play a key regulatory role in seed germination [32], vegetative growth [33], flowering transition [34], flower meristem and flower organ formation [35], flower development [36], seed and fruit development [37], senescence [38], and other development processes. *G. biloba* is a typical representative of gymnosperms, and there is great significance to studying its flowering mechanism. In this work, based on the important role of MADS-box in plan flower development, the members of MADS-box gene family were systematically analyzed in *G. biloba*. A total of 26 MADS-box genes were identified in *G. biloba*, and their basic physicochemical properties, chromosomal location, gene collinearity, conserved motifs, gene structure, phylogenetic relationship, miRNA target site, *cis*-acting elements, and expression profiles were studied in detail. The findings of this work can provide a theoretical basis for the future study of the mechanism and function of the MADS-box gene family in *G. biloba* (Figure S1).

## 2. Materials and Methods

### 2.1. Materials

The experimental material was the 35-year-old grafted *G. biloba* tree 'Jiafoshou', which was planted in the Ginkgo Germplasm Resource Nursery of Yangtze University, (N30.35, E112.14), China. The samples of early, metaphase, and later development of ovulate strobilus (OS1, OS2, and OS3) and early, metaphase, and later development of microstrobilus (M1, M2, and M3) were obtained from three female and three male plants, respectively (Figure S2). The sampling time was from March to May 2021. The samples were frozen in liquid nitrogen immediately after collection and stored in a refrigerator at $-80\,^\circ$C for the extraction of total RNA for qRT-PCR analysis.

### 2.2. Identification of MADS-Box Family Genes in G. biloba

Genome-wide data of *G. biloba* were obtained from the GigaDB website (http://gigadb.org/dataset/100613, accessed on 1 August 2021). The HMM profile of the MADS-box domain (PF00319) was downloaded from the Pfam protein family database (http://pfam.xfam.org/, accessed on 7 November 2021). The hmmsearch program (HMMER 3.0 package, http://hmmer.org/download.html, accessed on 7 November 2021) was employed against the whole protein sequence by using the HMM of the MDAS-box domain (PF00319) as the query file with E value $\leq 10^{-5}$. Through multiple alignment and checking the conserved domain on the SMART website (http://smart.embl-heidelberg.de/, accessed on 11 August 2021), repetitive sequences and sequences with incomplete conserved domains

were excluded, and the members of the MADS-box family in *G. biloba* were identified for further analysis. The physical and chemical properties of the GbMADS proteins were analyzed via the ExPASy website (http://web.expasy.org/protparam/, accessed on 12 August 2021). The online tool of WoLF PSORT (https://wolfpsort.hgc.jp/, accessed on 12 August 2021) was used to predict the subcellular localization.

### 2.3. Chromosomal Location and Collinearity Analyses for GbMADS Genes

The General Feature Format (GFF) was downloaded from the GigaDB (http://gigadb.org/dataset/100613, accessed on 1 August 2021), and the physical location of all *GbMADS* genes on the chromosome was drawn using the TBtools tool [39]. To unveil duplication patterns of each *GbMADS* gene in *G. biloba* and the genome of other species (*A. thaliana*, *Oryza sativa*, *P. trichocarpa*, and *Gnetum gnemon*), TBtools [39] was used to determine genomic collinearity with default parameters. The genome sequence files and gene structure annotation files of *A. thaliana*, *O. sativa*, *P. trichocarpa*, and *G. gnemon* are downloaded from NCBI (https://www.ncbi.nlm.nih.gov/, accessed on 7 November 2021).

### 2.4. Conserved Motifs, and Gene Structure Analysis

The evolutionary tree of members of the GbMADS gene family was constructed using ClustalX2 (http://www.clustal.org/clustal2/, accessed on 10 September 2021) implemented in MEGA 7.0 (https://www.megasoftware.net/download_form, accessed on 7 November 2021) with default parameters. The MEME online website (https://meme-suite.org/meme/, accessed on 10 September 2021) was used to analyze the conserved protein domains. The maximum number of conserved domains was set to 10, and the rest were set as default parameters. Visualize the results of conservative protein domain with TBtools [39]. The exon–intron structures of *GbMADS* genes were mapped by TBtools [39].

### 2.5. Phylogenetic Analysis and Classification of the GbMADS Genes

The MADS-box family protein sequences of *A. thaliana*, *Salix suchowensis*, *O. sativa*, *Pinus radiata*, *G. gnemon*, *Picea abies*, *Cryptomeria japonica*, *Amborella trichopoda*, and *G. biloba* were sequenced by ClustalX2 (http://www.clustal.org/clustal2/, accessed on 1 November 2021). Using MEGA 7.0 software (https://www.megasoftware.net/download_form, accessed on 1 November 2021), the Neighbor-Joining (NJ) method was used to construct a phylogenetic tree with 1000 bootstrap replicates. The MADS-box family protein sequences of *A. thaliana* were downloaded from the TAIR (https://www.arabidopsis.org/, accessed on 1 October 2021). The MADS-box gene accession numbers and loci used information of *S. suchowensis*, *O. sativa*, *P. radiata*, *G. gnemon*, *P. abies*, *C. japonica*, and *A. trichopoda* were found from the literature [40–44], and the protein sequences were downloaded from NCBI (https://www.ncbi.nlm.nih.gov/, accessed on 20 October 2021). The evolutionary tree was further embellished by using the EvolView v2 online website (https://evolgenius.info/evolview-v2/#login, accessed on 2 November 2021).

### 2.6. miRNA Target Prediction

In the early stage, a small RNA sequencing database was constructed and the miRNA target genes were predicted used the TargetFinder software (v1.6, http://carringtonlab.org/resources/targetfinder, accessed on 1 February 2020) with the Parameter of -c $^3$ to obtain miRNA-mRNA interaction pairs [3,45]. In this study, combined with the miRNA-mRNA interaction database, the miRNA targeting-related MADS genes ($R > 0.89$, $p < 0.01$) were identified, and the regulatory relationship through OmicShare Tools (https://www.omicshare.com/tools/Home/Soft/cytoscape, accessed on 1 December 2021) were displayed.

### 2.7. Promoter Analysis of the GbMADS Genes

The promoter sequence of the *GbMADS* genes (2000 bp upstream of ATG) was extracted through the TBtools [39] to predict the cis-acting elements via the PlantCARE

(http://bioinformatics.psb.ugent.be/webtools/plantcare/html/, accessed on 1 January 2022). TBtools [39] was used for visual analysis.

### 2.8. Gene Expression Analysis

The RNA was extracted with a TaKaRa MiniBEST Plant RNA Extraction Kit. The first strand of cDNA was synthesized with a HiScript® II Q RT SuperMix for qPCR (+gDNA wiper) kit (TaKaRa, Dalian, China). Primers were designed using the Integrated DNA Technologies online website (https://sg.idtdna.com, accessed on 7 November 2021). The primer sequence (Table S1) was submitted to Sangon Biotech for synthesis. The qRT-PCR was performed using the ChamQ Universal SYBR qPCR Master Mix enzyme system (Vazyme, Nanjing, China) and the internal reference gene was *GbGAPDH* [46]. The relative expression values of *GbMADS* genes were analyzed using the $2^{-\Delta\Delta Ct}$ method, and the data were technically replicated with error bars representing the mean $\pm$ SD ($n$ = 3). Excel 2019 (Microsoft Corp., Redmond, WA, USA) was used to process the experimental data, and SPSS 22.0 software (IBM Corp., Armonk, NY, USA) was used for one-way analysis of variance ($p < 0.05$). TBtools [39] was used to draw the gene expression heatmap.

## 3. Results

### 3.1. Identification and Physicochemical Properties of the MADS-Box Gene Family in G. biloba

A total of 26 MADS-box genes were identified in *G. biloba*, which were named *GbMADS1–GbMADS26* (Table 1). The results showed that the length of the CDS sequence of *GbMADS* was 276 bp (*GbMADS8*)~1338 bp (*GbMADS21*), the number of amino acids were 91~445 aa, and the molecular weight (MW) of the proteins were 10.58~50.95 kDa. The isoelectric points (PI) ranged from 4.96 to 10.20, of which eighteen proteins were alkaline and eight proteins were acidic. The grand averages of hydropathicity (GRAVY) were in the range of −0.917 to 0.131. Only GbMADS22 is a non-hydrophilic protein, while the others are hydrophilic proteins. Among GbMADS proteins, only GbMADS3, GbMADS8, and GbMADS22 with an instability index (II) less than 40, and the rest were greater than 40, indicating that the GbMADS proteins are an unstable protein. Most proteins were located in the nucleus, only GbMADS3 was located in the extracellular matrix, and GbMADS11 and GbMADS13 were located in the chloroplast.

**Table 1.** Basic information of MADS-box family members in *G. biloba*.

| Gene Name | Gene ID | Chromosome Location | CDS (bp) | Protein Length (aa) | MW (KDa) | GRAVY | pI | II | AI | Subcellular Localization |
|-----------|---------|---------------------|----------|---------------------|----------|-------|-----|------|-------|--------------------------|
| *GbMADS1* | Gb_01884 | Chr1: 713,430,121 . . . 713889578 | 762 | 253 | 28.57 | −0.521 | 9.8 | 45.66 | 87.11 | Nucleus |
| *GbMADS2* | Gb_03068 | Chr7: 279,659,904 . . . 279661933 | 489 | 162 | 18.42 | −0.608 | 9.96 | 58.83 | 78.27 | Nucleus |
| *GbMADS3* | Gb_03807 | Chr7: 186,875,289 . . . 186876549 | 573 | 190 | 20.98 | −0.104 | 10.05 | 32.83 | 88.26 | Extracell |
| *GbMADS4* | Gb_05128 | Chr7: 190,259,000 . . . 190423917 | 735 | 244 | 28.05 | −0.668 | 4.96 | 44.36 | 86.27 | Nucleus |
| *GbMADS5* | Gb_05359 | Chr3: 225,287,517 . . . 225289874 | 1320 | 327 | 37.65 | −0.738 | 5.31 | 64.01 | 71.87 | Nucleus |
| *GbMADS6* | Gb_12581 | Scaffold 14912: 35,255 . . . 35785 | 369 | 122 | 13.74 | −0.397 | 10.02 | 40.68 | 83.11 | Nucleus |
| *GbMADS7* | Gb_12586 | Chr9: 262,143,415 . . . 262144485 | 1071 | 356 | 40.51 | −0.65 | 6.32 | 60.9 | 71.8 | Nucleus |
| *GbMADS8* | Gb_12778 | Chr6: 294,934,960 . . . 295095100 | 276 | 91 | 10.58 | −0.693 | 9.76 | 35.09 | 70.77 | Nucleus |
| *GbMADS9* | Gb_15398 | Chr12: 279,825,002 . . . 279826961 | 834 | 277 | 31.89 | −0.405 | 8.86 | 40.93 | 81.66 | Nucleus |
| *GbMADS10* | Gb_16301 | Chr7: 366,097,174 . . . 366139501 | 666 | 221 | 25.38 | −0.605 | 8.96 | 60.62 | 85.25 | Nucleus |
| *GbMADS11* | Gb_19178 | Chr1: 717,036,617 . . . 717203413 | 741 | 246 | 28.20 | −0.594 | 6.26 | 59.86 | 84.31 | Chloroplast |
| *GbMADS12* | Gb_19258 | Chr6: 423,032,522 . . . 423033532 | 1011 | 336 | 37.20 | −0.704 | 8.33 | 62.85 | 67.71 | Nucleus |
| *GbMADS13* | Gb_21526 | Chr11: 610,402,766 . . . 610403275 | 510 | 169 | 19.41 | −0.499 | 10.17 | 53.10 | 79.59 | Chloroplast |
| *GbMADS14* | Gb_28587 | Chr2: 383,363,358 . . . 383365563 | 774 | 257 | 29.39 | −0.416 | 9.90 | 51.03 | 98.68 | Nucleus |
| *GbMADS15* | Gb_30604 | Chr1: 709,778,670 . . . 709874204 | 501 | 166 | 19.37 | −0.710 | 10.20 | 55.36 | 86.39 | Nucleus |
| *GbMADS16* | Gb_31417 | Chr1: 318,984,240 . . . 319003647 | 1146 | 381 | 43.30 | −0.672 | 5.96 | 47.31 | 73.02 | Nucleus |
| *GbMADS17* | Gb_33168 | Chr3: 242,930,058 . . . 242930687 | 630 | 209 | 24.35 | −0.708 | 5.53 | 63.85 | 80.19 | Nucleus |
| *GbMADS18* | Gb_36364 | Chr1: 707,044,181 . . . 707202195 | 759 | 252 | 29.17 | −0.917 | 8.88 | 54.23 | 77.30 | Nucleus |
| *GbMADS19* | Gb_37613 | Scaffold 11131: 15,283 . . . 16293 | 1011 | 336 | 37.28 | −0.680 | 8.33 | 58.81 | 67.11 | Nucleus |
| *GbMADS20* | Gb_38365 | Chr1: 560,863,035 . . . 560864326 | 405 | 134 | 15.27 | −0.480 | 9.34 | 50.94 | 72.09 | Nucleus |
| *GbMADS21* | Gb_38883 | Chr1: 876,632,006 . . . 876634445 | 1338 | 445 | 50.95 | −0.667 | 5.93 | 47.36 | 69.69 | Nucleus |
| *GbMADS22* | Gb_38922 | Chr1: 710,260,574 . . . 710261629 | 498 | 165 | 18.37 | 0.131 | 9.37 | 38.36 | 96.42 | Nucleus |
| *GbMADS23* | Gb_39109 | Chr1: 760,814,262 . . . 760970131 | 600 | 199 | 22.58 | −0.513 | 6.22 | 66.05 | 90.15 | Nucleus |
| *GbMADS24* | Gb_40092 | Scaffold 1360: 32,384 . . . 33394 | 1011 | 336 | 37.26 | −0.687 | 8.33 | 59.06 | 66.55 | Nucleus |
| *GbMADS25* | Gb_41549 | Chr1: 705,678,899 . . . 705806511 | 738 | 245 | 28.35 | −0.754 | 9.61 | 46.67 | 82.37 | Nucleus |
| *GbMADS26* | Gb_41550 | Chr1: 705,910,701 . . . 705911828 | 606 | 201 | 22.54 | −0.051 | 9.28 | 59.08 | 100.40 | Nucleus |

### 3.2. Chromosome Mapping and Collinearity Analysis of GbMADS

As shown in Figure 1, *GbMADS* genes were distributed on eight chromosomes, with 11 genes on chromosome 1, followed by four genes on chromosome 7. There were two genes on each of chromosomes 3 and 6. There was only one gene on chromosomes 2, 9, 11, and 12. The remaining three *GbMADS* genes were located in scaffold 14912, scaffold 11131, and scaffold 1360.

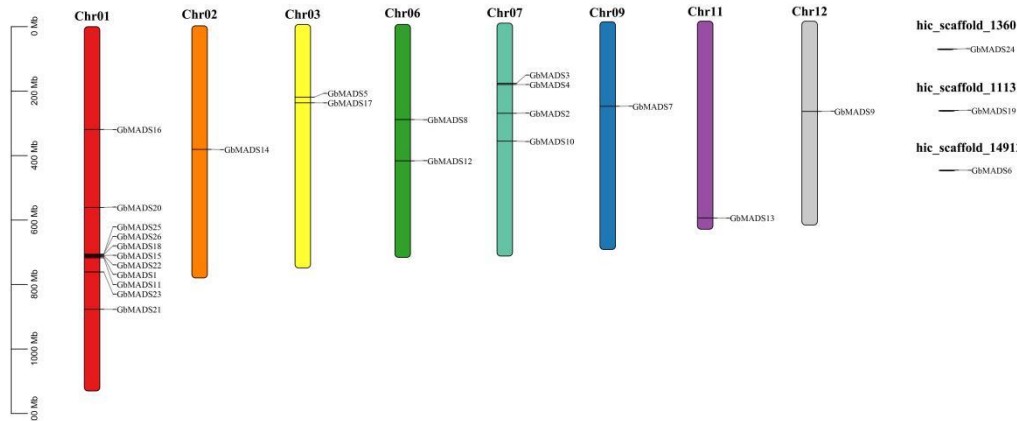

**Figure 1.** Chromosome location of GbMADS gene family in *G. biloba*. Each chromosome is marked with different colors. The length of chromosomes can be estimated by the scale on the side.

To study the homology of the genes, intra-species collinearity analysis of the *GbMADS* genes was carried out. At the same time, species collinearity analysis was performed on the MADS-box family of *G. biloba*, *A. thaliana*, *O. sativa*, *P. trichocarpa*, and *G. gnemon* (Figure S3). The results showed that there was no collinearity among the 26 *GbMADS* genes. Except *P. trichocarpa*, the gene collinearity between *G. biloba* and other species was not close or had no collinearity. No collinearity was found between *GbMADS* genes and MADS-box genes in other species.

### 3.3. Classification, Conserved Motifs, and Gene Structure Analysis of the GbMADS Family

The phylogenetic tree was constructed using the GbMADS protein sequence. The results showed that GbMADS were divided into type-I (yellow part) and type-II (blue part) (Figure 2). Ten conserved motifs in the GbMADS gene family were predicted by the online software MEME, which were named motif1–motif10 (Table S2). Motif 1 and motif 5 are MADS domains, and motif 2 is a K-box domain. As shown in Figure 2, 10 type-I GbMADS proteins only possessed a MADS domain and no K-box domain. Among the 16 type-II GbMADS proteins, except for the K-box domain deletion of GbMADS6, GbMADS8, GbMADS20, GbMADS22, and GbMADS26, the other type-II GbMADS proteins contained MADS and K-box domains. In addition, GbMADS5, GbMADS17, and GbMADS21 in the same branch of the type-I gene shared three motifs, and GbMADS12, GbMADS19, and GbMADS24 shared the same seven motifs, indicating that the conserved motifs of closely related proteins are similar, suggesting that they may have similar functions.

Gene structure analysis indicated that the number and distribution of exons and gene length of type-I (yellow part) and type-II (blue part) *GbMADS* genes were quite different (Figure 3). However, the most closely related members of the same subfamily possessed similar gene structures in terms of exon number or intron length. The similarity of gene structure was consistent with the phylogenetic analysis. The gene length of the type-I genes was short, and the number of exons was significantly less than that of the type-II gene. On the contrary, the gene length of the type-II genes varied greatly, and the number of exons was large and unevenly distributed. The number of exons of *GbMADS* genes varied from 1 to 11. Among them, *GbMADS7, 12, 13, 17, 19*, and *GbMADS24* possessed only one exon,

which belongs to a type-I gene. *GbMADS1* and *GbMADS16* possessed 11 exons, which belong to type-II genes.

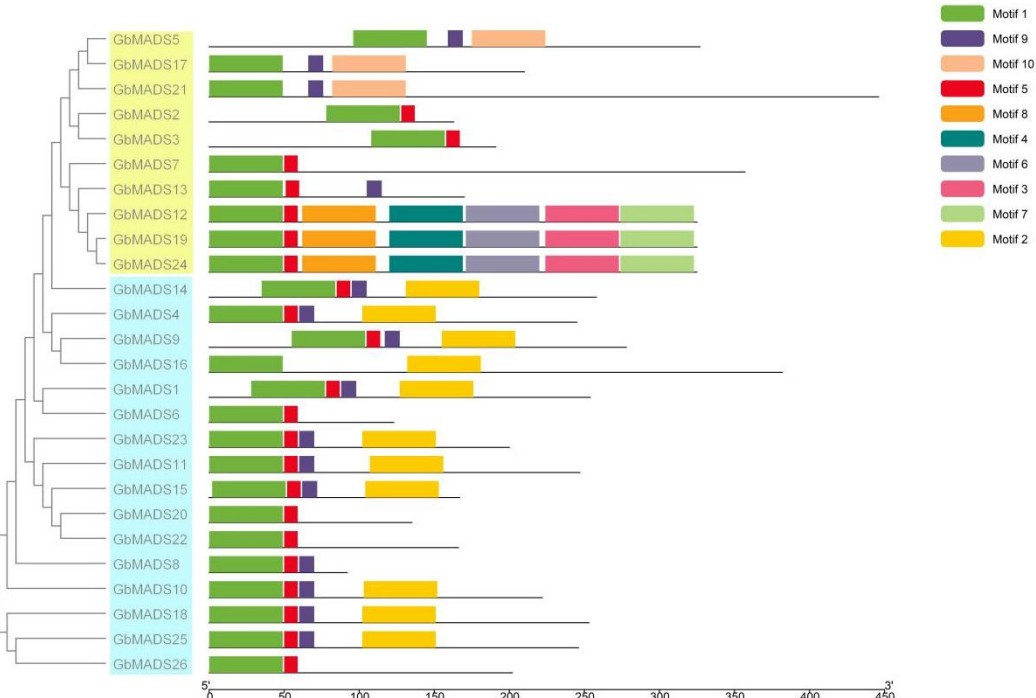

**Figure 2.** Conserved motif of GbMADS gene family in *G. biloba*. In the phylogenetic tree constructed based on the full-length sequence of GbMADS proteins, the yellow background color was labeled with type-I GbMADS proteins, and the blue background color was labeled with type-II GbMADS proteins. Boxes of different colors represent 10 putative motifs. The length of protein can be estimated using the scale at the bottom.

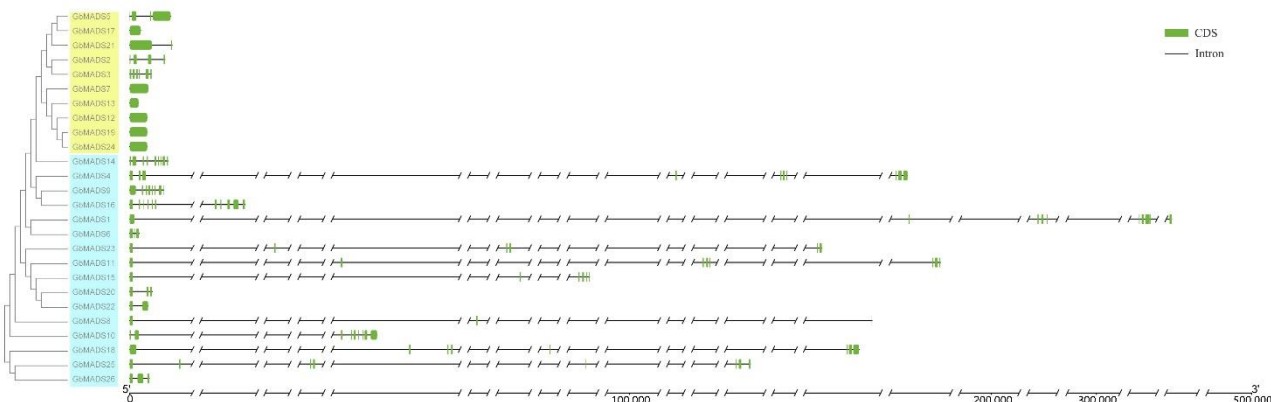

**Figure 3.** Gene structures of GbMADS gene family in *G. biloba*. In the phylogenetic tree constructed based on the full-length sequence of GbMADS proteins, the yellow background color was labeled with type-I GbMADS proteins, and the blue background color was labeled with type-II GbMADS proteins. CDS and intron were represented by green boxes and black lines, respectively. Because the length of some *GbMADS* genes is too long, the intron region is partially cut, and the cut part is separated by two slashes.

### 3.4. Phylogenetic Analysis of the GbMADS Gene Family

Ten type-I GbMADS protein sequences were phylogenetically analyzed with the type-I MADS-box proteins of *A. thaliana*, *O. sativa*, and *S. suchowensis*. As shown in Figure 4a, five were found to belong to the Mα subclass, three to the Mβ subclass, and two to the Mγ subclass. Sixteen type-II GbMADS protein sequences were phylogenetically analyzed

with the type-II MADS-box protein sequences of *A. thaliana*, *S. suchowensis*, *O. sativa*, *G. gnemon*, *C. japonica*, *P. abies*, *A. trichopoda*, and *P. radiata*. The results showed (Figure 4b) that one *GbMADS* gene (*GbMADS16*) was assigned to the MIKC * branch and fifteen *GbMADS* genes were assigned to the MIKC^C branch. There was one *GbMADS* gene in each subfamily of *TM3*, *AG*, *AGL6*, *STMADS11*, and *GGM13*. Furthermore, *GbMADS14* and *AGL12* are sister branches, *GbMADS25*, *GbMADS26*, and *AGL2* are sister branches, and seven genes such as *GbMADS1* are along the same branch as the two subfamilies of *AGL2* and *AGL6*. *GbMADS* was not found in the *AP1*, *FLC*, *AGL15*, *AGL17*, *TM8*, and *AP3/PI* subfamilies.

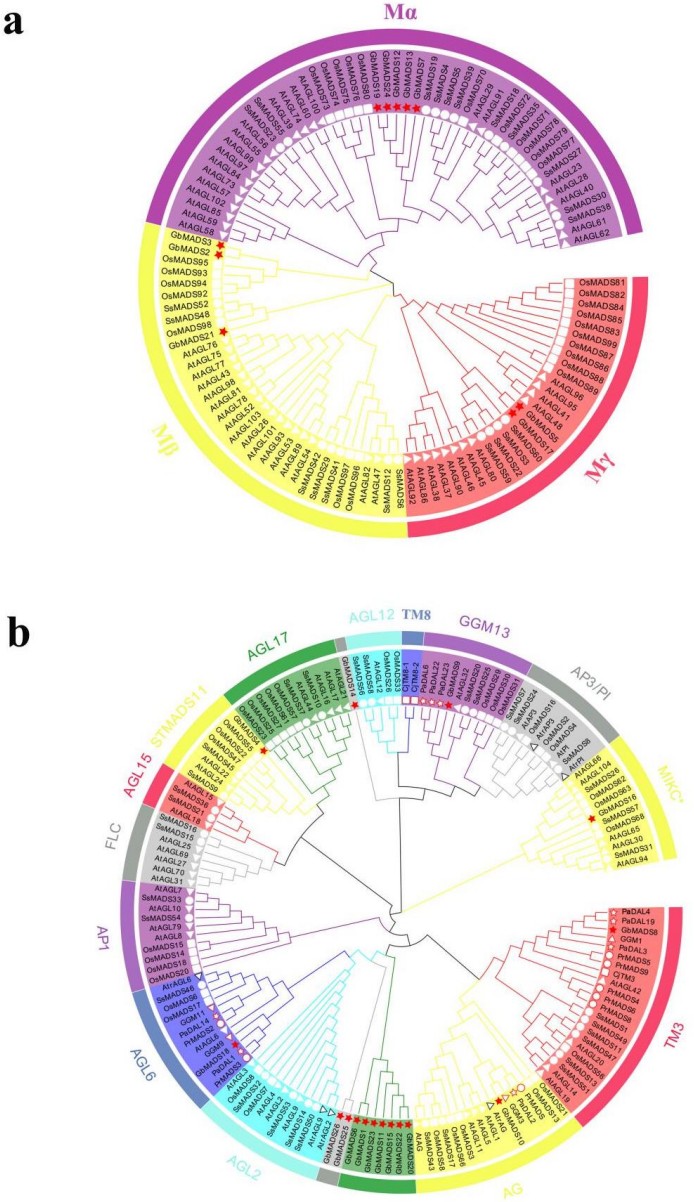

**Figure 4.** (**a**) Evolutionary tree of type-I MADS-box gene family; (**b**) Evolutionary tree of type-II MADS-box gene family. The phylogenetic tree was generated with 1,000 bootstrap replicates using the neighbor-joining (NJ) method. Each subgroup's name was in the outer circle of the phylogenetic tree, and different subgroups were marked with different colors. Genes of different species were marked in the inner circle with different shapes and colors, Gb, *Ginkgo biloba* (star, red, red border); At, *Arabidopsis thaliana* (triangle, white, white border); Os, *Oryza sativa* (square, white, white border); Ss, *Salix suchowensis* (circular, white, white border); GGM, *Gnetum gnemon* (triangle, white, red border); Cj, *Cryptomeria japonica* (square, white, red border); Pa, *Picea abies* (star, white, red border); Atr, *Amborella trichopoda* (triangle, white, black border); Pr, *pinus radiata* (circular, white, red border).

### 3.5. Prediction of the miRNA Target Sites of the GbMADS Gene Family

miRNAs are important hubs for regulating gene transcription and expression. Analyzing the miRNA target sites in *GbMADS* genes is of great significance for understanding gene function. In this study, we found miRNA targets sites in 16 *GbMADS* genes whose expression might be regulated by miRNAs (Table S3, Figure 5). Among them, novel_miR_1559 (MIR5067 family member) might target *GbMADS7*, novel_miR_322 (MIR396 family member) might target *GbMADS15* and *GbMADS20*, novel_miR_1291 (MIR7717 family member) might target *GbMADS25*, and some other newly discovered miRNAs might target *GbMADS* genes to participate in the regulatory process in cells. In addition, this study also found that there can be multiple miRNA target sites on a gene, and the same miRNA can regulate different target genes. This confirms that miRNA has a complex regulatory network in plant cells and can participate in a variety of growth and development processes, such as hormone and immune response, flower organ regulation, male and female flower differentiation, and response to biological and abiotic stress [47].

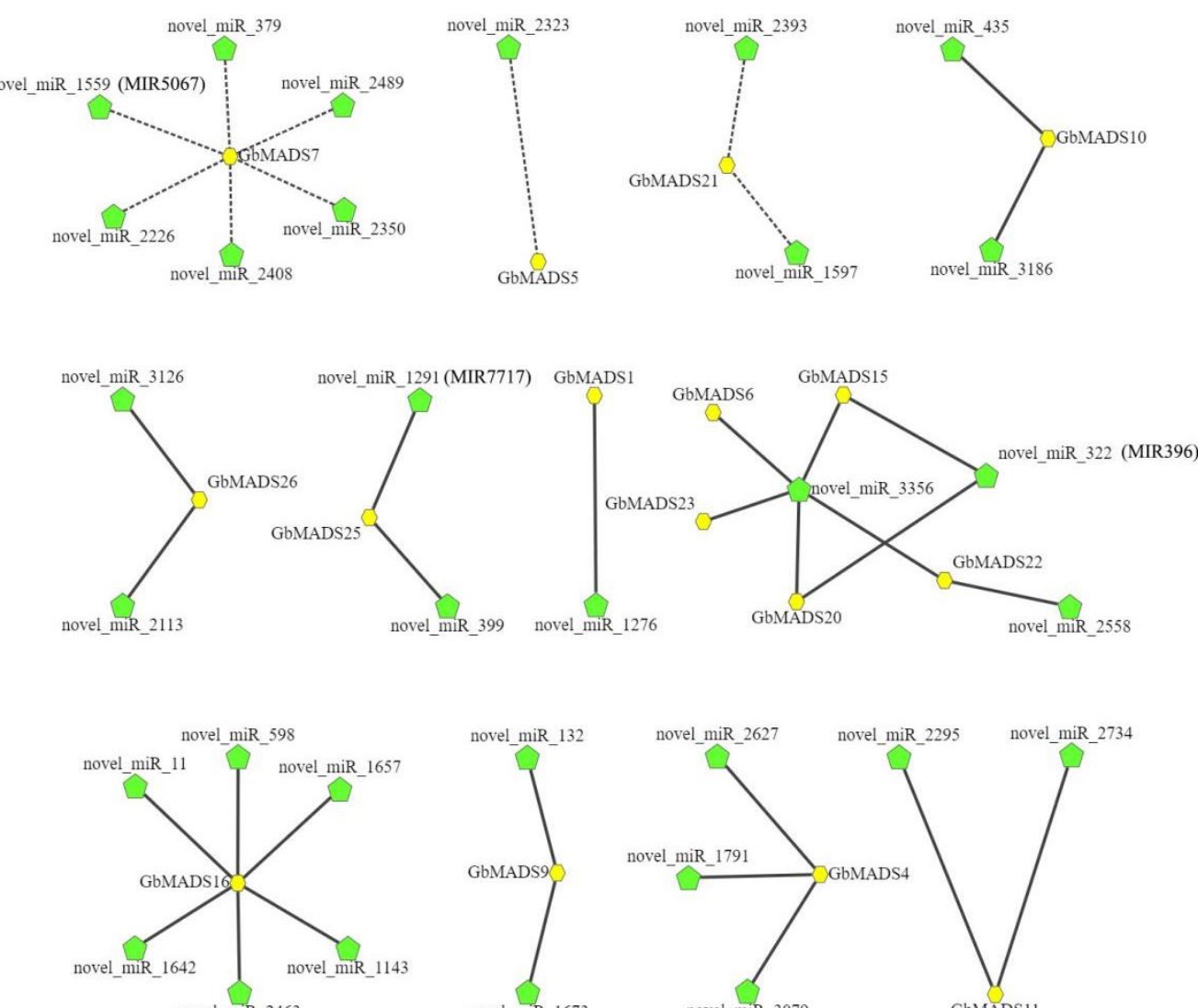

**Figure 5.** miRNA-directed network diagram of *GbMADS* genes in *G. biloba*. The yellow hexagon represents *GbMADS* genes, and the green Pentagon represents miRNA.

### 3.6. Promoter Sequence Analysis of the GbMADS Gene Family

To further study the potential function of GbMADS family members, the upstream promoter elements of the *GbMADS* genes were predicted. As shown in Figure 6, these

promoter sequences not only contained basic elements such as initiation transcription site and TATA-box, but also contained many cis-acting elements related to gene function, such as stress-response elements (TC-rich repeats and ARE), hormone-response elements (ABRE, P-box, and TCA-element), light-response elements (AE-box, AT1-motif, ATC-motif, Box 4, G-Box, GA-motif, GATA-motif, I-box, MRE, chs-CMA2a, and TCT-motif), and the gene expression and physiological regulatory element (GCN4-motif). Among them, the number of optical response elements was the largest. The flower development process of *G. biloba* was found to be closely related to photoperiod, hormone, low temperature, and meristem. It is evident that *GbMADS* genes play an important regulatory role in this process.

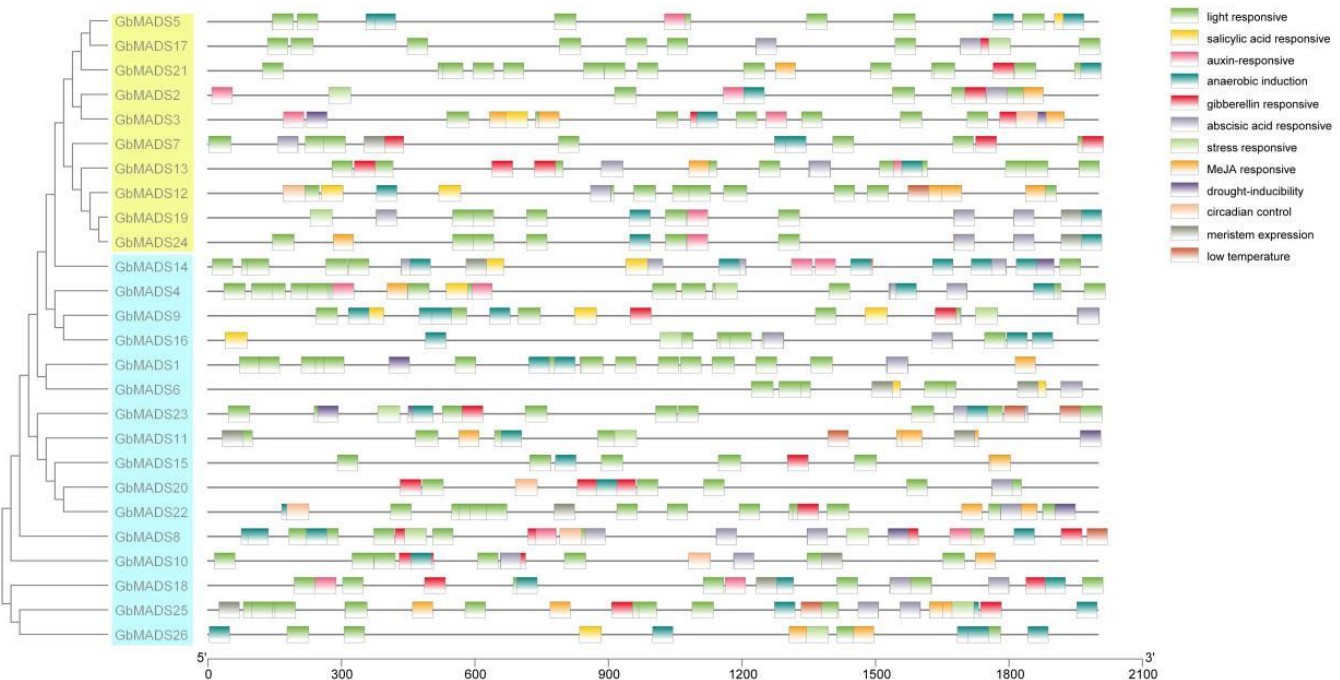

**Figure 6.** Promoter cis-acting elements of GbMADS gene family in *G. biloba*. In the phylogenetic tree constructed based on the full-length sequence of GbMADS proteins, the yellow background color was labeled with type-I GbMADS proteins, and the blue background color was labeled with type-II GbMADS proteins. Boxes of different colors represent different response elements.

*3.7. Expression Profile Analysis of the GbMADS Genes in the Male and Female Flowers of G. biloba*

The expression patterns of the *GbMADS* genes in six stages of OS1, OS2, OS3, M1, M2, and M3 were shown in Figure S4. The results can be visualized as heat maps to show that *GbMADS* genes were expressed to varying degrees at different stages of flower development (Figure 7). *GbMADS9* and *GbMADS17* were highly expressed in the M2 phase, and *GbMADS16* was much more highly expressed in the M1 and M2 phase than the other genes. As a member of the MIKC * branch in type-II, *GbMADS16* may play a key role in male flower development. Except for *GbMADS2* and *GbMADS7*, the expression of the type-I *GbMADS* genes (yellow part) in the male flowers was higher than that in the female flowers. With the exception of *GbMADS9*, *GbMADS14*, and *GbMADS16*, the expression of other type-II *GbMADS* genes in the female flowers was higher than that in the male flowers. These results show that there are differences in the expression level of *GbMADS* genes in the same gene family and even in the same subfamily.

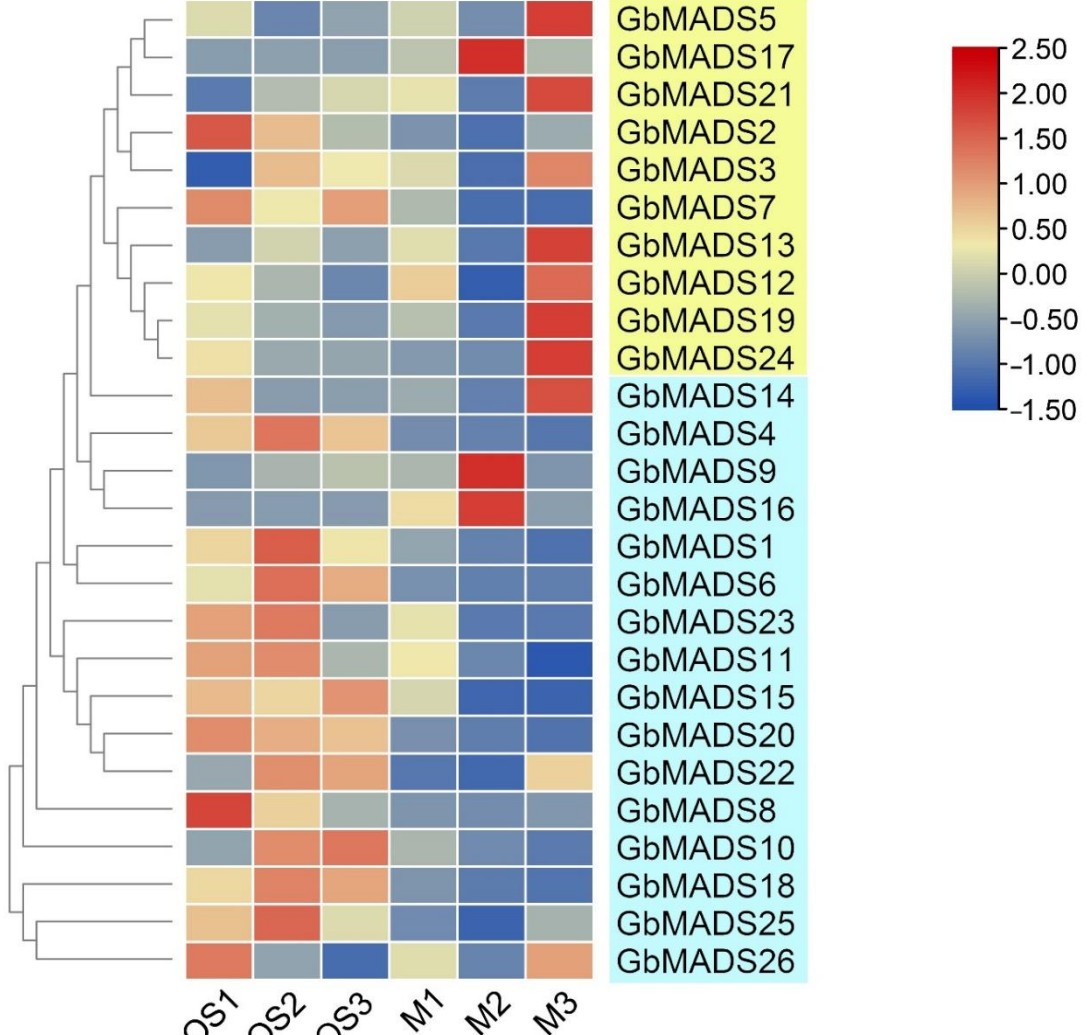

**Figure 7.** Heatmap representation of *GbMADS* genes in male and female flowers of *G. biloba*. The OS1, OS2, OS3, M1, M2, and M3 represent early development of ovulate strobilus, metaphase development of ovulate strobilus, later development of ovulate strobilus, early development of microstrobilus, metaphase development of microstrobilus, and later development of microstrobilus, respectively. The colored scale for the different expression levels was shown to the top right.

## 4. Discussion

### 4.1. Genealogical Evolution of the GbMADS Genes in G. biloba

The flower organ is an important basis for studying plant evolution and classification, and the study of the floral organ has always been a research hotspot. Molecular studies show that flower development is regulated by multiple transcription factors, including the MADS-box family, MYB family, and LEAFY family, in which MADS-box transcription factors play a key role. The number and types of MADS-box genes differ in different species. The number of MADS-box genes in angiosperms such as *A. thaliana* (107) [26] and tomato (131) [48], and in gymnosperms such as *C. japonica* (12) [44] and *Cunninghamia lanceolata* (47) [49]. Some species have few or no type-I genes, such as *Marchantia polymorpha*, *Jatropha curcas*, and *Picea glauca*, but up to 271 type-I genes have been found in *Camelina sativa*. Type-II genes are few or absent in some species, such as *Chlamydomonas reinhardtii*, *P. abies*, and *Selaginella moellendorffii*, but 209 type-II genes are found in soybean [50,51]. The number of different evolutionary groups in different species differs, demonstrating differences in pedigree-specific birth and death patterns, which may reflect the adaptive response of plants to selection pressure to increase their adaptability to the environment.

The size and complexity of gymnosperm genomes present a great challenge to the study of the evolutionary relationships of gymnosperm lineages. So far, the whole-genome sequence of gymnosperms is limited to a few conifers (*Picea asperata*, *Pinus taeda*, and *Pinus lambertiana*, etc.), *G. biloba,* and Gnetophyta, and there are some contradictions in the phylogenetic evidence among these species, which hinders our understanding of the genome evolution of all seed plants. *G. biloba* is the sole extant representative of the gymnosperm family Ginkgoaceae. The collinearity analysis results (Figure S3) showed that 26 *GbMADS* genes exhibited no collinearity. The evolutionary homology between *G. biloba* gene and herbs (*A. thaliana*, *O. sativa*) and other gymnosperm genes was low, and the evolutionary relationship with woody plants (*P. trichocarpa*) was relatively close, whereas no collinearity of MADS-box genes was detected in these species. The evolutionary processes of the *GbMADS* genes remain to be further studied.

### 4.2. Bioinformatics Prediction of GbMADS Gene's Function

In this study, the GbMADS gene family was identified genome-wide, and 26 *GbMADS* genes were divided into 11 subfamilies (Figure 4). Compared with *A. thaliana*, the number of *GbMADS* genes was less, and no *GbMADS* genes were found in *AP1*, *FLC*, *AP3/PI* and other subfamilies. In addition, the genome size of *G. biloba* (10.61 Gb) was more than 80 times that of *A. thaliana* (119.67 Mb) and much larger than most other sequenced plant species. The above results showed that there was no significant correlation between genome size and gene number. Guan et al. showed that *G. biloba* has experienced genome-wide replication and a large number of long terminal replication (LTR) insertion events, resulting in the repeat sequences accounting for more than three-quarters of the whole genome [52]. The activity of transposable elements makes the average intron length of *G. biloba* longer than that of the other sequenced species. Combined with the above research, the structure and conserved motif of each subfamily gene were further analyzed according to the classification results of the phylogenetic tree analysis (Figures 2 and 3). A significant difference in the gene structure of GbMADS was found. The gene length of the type-I *GbMADS* genes was shorter, the number of exons was significantly less than that of the type-II gene, and the intron length was shorter. The gene length of the type-II gene varied greatly, the number of exons was large and unevenly distributed, and the length of the introns was long. Previous studies have shown that MIKC$^{\text{C}}$ genes (*AP1/CAL/FUL*, *AGL2*, etc.) with a redundant gene structure may have the function of controlling very obvious phenotypic effects in plants [53,54].

As a gymnosperm, *G. biloba* has no specific typical flower structure, and the flower organ identity-related gene *AP1*, which contributes to the formation of the sepals and petals of angiosperms, may have been lost in the process of evolution. *FLC* is the central regulator of vernalization-induced flowering [55]. The absence of the FLC subfamily indicates that vernalization is not necessary for the flowering in *G. biloba* [56]. After low-temperature vernalization, a higher ambient temperature is conducive to the accumulation of organic matter in *G. biloba*, so as to promote flowering. Prolonged low temperature inhibits the flowering of *G. biloba*. By contrast, the *AGL12* gene has been found in *G. biloba*, *C. lanceolata* [49], *P. taeda* [57], *A. thaliana* [58], and *O. sativa* [41], suggesting that the gene may be functionally conserved and important for the development of the flowers and cones of angiosperms and gymnosperms. At the same time, several *GbMADS* genes close to Class E genes (*AGL2* and *AGL6*) are identified in *G. biloba*. Most Class E genes are related to the organ recognition of the petals, stamens, and carpels.

Many studies have found that miRNAs can target MADS-box gene family members to participate in plant flower development. In pear, miR6390 and *Dormancy-associated MADS-box genes* (*DAMs*) jointly control the dormancy transition of flower buds [59]. miR5179 plays an important role in the diversification of the organs of the perianth in orchids through the inhibitory regulation of the *DEF-like* gene [60]. In *Catharanthus roseus*, miR396 targets *SVP* gene to inhibit flowering and may participate in the formation of abnormal flowers [61]. MiR396 acts on *VERNALIZAITON1* (*VRN1*) and participates in the vernalization process

of grape flowers together with other regulatory elements [62]. In this study, 16 *GbMADS* genes were found to have miRNA target sites. It was speculated that *GbMADS7* could be targeted by novel_miR_1559 (MIR5067 family member), *GbMADS15, 20* could be targeted by novel_miR_322 (MIR396 family member), and *GbMADS25* could be targeted by novel_miR_1291 (MIR7717 family member) to participate in the flower development of *G. biloba*. In addition, promoter region analysis found that most *cis*-elements in the *GbMADS* genes were related to light, hormone, temperature, and stress. These results suggest that the *GbMADS* genes may play an important role in flower development and differentiation, which is also supported by research results in *A. thaliana* [26].

*4.3. Expression Pattern and Function of the GbMADS Genes in the Male and Female Flowers of G. biloba*

As a typical dioecious plant, *G. biloba* blooms asynchronously, which can effectively prevent selfing, reduce inbreeding in morphology, and promote genetic diversity. Combined with the phylogenetic tree (Figure 4), the gene expression levels of *GbMADS* genes in *G. biloba* flowers were analyzed (Figure S4, Figure 7). Most type-I genes were found to be more expressed in the M3 stage than that in the other tissues. Except for *GbMADS9*, *GbMADS14*, and *GbMADS16*, the expression of other type-II genes in the female flowers was higher than that in the male flowers. In addition, it was found that the expression profiles of genes on the same branch were similar. For example, the expression levels of *GbMADS19, 24, 12*, and *13* on the Mα branch were higher in the M3 stage, followed by the OS1 period. In this study, the expression of *GbMADS2* on the Mβ branch and *GbMADS8* on the TM3 branch was high in the OS1 stage, and on the contrary, the expression was very low in the M1 stage, indicating that these two genes may play an important role in the differentiation of the male and female flower buds. Interestingly, we also found that the expression of the only MIKC * gene *GbMADS16* in the male and female flowers varied greatly, especially in the M2 period, followed by the M1 stage. On the contrary, its expression did not change significantly in different stages of female flower development. It is speculated that this gene plays a key regulatory role in the process of male flower development.

**5. Conclusions**

In this study, a total of 26 *GbMADS* genes distributed in 11 subfamilies were identified. They are unevenly distributed on chromosomes, with different lengths, different gene structures, and different basic physicochemical properties of their encoded proteins. However, the distribution of conserved motifs and cis-acting elements of GbMADS on the same branch or near source branches is similar, indicating that the functions of the *GbMADS* genes are different, but that the gene functions in the same subfamily are relatively conserved. In addition, we also found that some *GbMADS* genes had miRNA target sites, suggesting that the *GbMADS* genes and miRNAs might regulate flower development by acting on structural genes. The qRT-PCR results showed that *GbMADS* genes were differentially expressed in the male and female flowers of *G. biloba* at different developmental stages. Most type-I and type-II genes were highly expressed in the male and female flowers, respectively, and the expression patterns of genes on the same branch were similar. Among them, the only MIKC * gene *GbMADS16* has the highest expression in the M2 stage. The above results indicated that different *GbMADS* genes were expressed differently in male and female flowers of *G. biloba*. It is speculated that MADS-box genes play an important role in flowering regulation and flower organ development of *G. biloba*. In the future, based on the results of this study, we will further study the regulatory relationship between the key *GbMADS* genes, miRNAs, and other flower development genes through gene cloning, genetic transformation, tobacco transient co-expression, yeast hybridization, and other genetic engineering technologies. So as to improve the flower development regulatory network of *G. biloba*.

**Supplementary Materials:** The following supporting information can be downloaded at: https://www.mdpi.com/article/10.3390/f13111953/s1, Figure S1: Methods framework; Figure S2: Image of male and female flowers of *G. biloba*. Figure S3: Collinearity analysis of MADS-box between *G. biloba* and other four plant species (*A. thaliana*, *O. sativa*, *P. trichocarpa*, *G. gnemon*). Gray lines in the background indicated the collinear blocks within *G. biloba* and other plant genomes, while red lines highlighted the syntenic MADS-box gene pairs; Figure S4: The expression level of 26 *GbMADS* genes in male and female flowers of *G. biloba*. (a) type-I genes; (b) type-II genes; All of the data shown reflect the average mean of three biological replicates ($n = 3$). Different letters in each figure represent a significant difference at $p < 0.05$; Table S1: qRT-PCR primer sequences of *GbMADS* genes; Table S2: Conserved motif sequence of GbMADS gene family in *G.biloba*; Table S3: List of relationship between *GbMADS* gene and miRNA.

**Author Contributions:** Conceptualization, J.Y. and F.X.; Methodology, X.C. and Z.L.; Software, W.Z.; Validation, K.Y., X.Z. and X.Y.; Formal Analysis, K.Y.; Writing—Original Draft Preparation, K.Y.; Writing—Review and Editing, J.Y., Y.L. and Q.W. All authors have read and agreed to the published version of the manuscript.

**Funding:** This work was supported by the National Natural Science Foundation of China, grant number 31670608.

**Data Availability Statement:** The data and results are available to every reader upon reasonable request.

**Conflicts of Interest:** The authors declare no conflict of interest.

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
