# Peer review of "Genome-Wide Identification and Expression Analysis of the MADS-Box Family in Ginkgo biloba"

_forests, doi:10.3390/f13111953_

Round 1

Reviewer 1 Report

This manuscript focused on the MADS-box family in Ginkgo biloba. Some of the results will provide important information for the diagnosis of nutrient deficiencies in lettuce. However, this manuscript needs further revision. Some suggestions were proposed as follows and I hope some of them would be helpful.

 1.     This manuscript (Ms) contains misprints and mistakes in English grammar and writing style. I recommend that the authors should use some help from a native English speaker or send the Ms to an English Editing Service that proofreads scientific writing.

2.     In the part of Materials and methods. Authors should provide detailed information about the data analysis process or relevant important parameters of some softwares. When you used the softwares (such as HMMER3.1, TBtools, Clustal X2, and so on.) to perform some analyses in the manuscript, you should cite relevant references or download websites.

3.     In the part of Materials and methods. You should show us more detailed information about the miRNA data analysis process. Where was the original miRNA data? Which software and relevant parameters you used? Which reference genome you used?

4.     In the part of Materials and methods. How to get the MADS-box family protein sequences of A. thaliana, Salix suchowensis, Oryza sativa, Pinus radiata, Gnetum gnemon, Picea abies, Cryptomeria japonica, Amborella trichopoda?

5.     Please provide more annotations and descriptions to clarify the meanings of some figures in the part of the results, which will help readers get a better understanding.

6.     Some of the formats of references were incorrect (such as in lines 441, 513, 520…). Some of the names of journals were abbreviations, but a few of them were full names. Check carefully and keep the format consistent according to the requirement of the journal.  

7.     Although the gene structure characteristic, evolution mechanism and co-expression network were analyzed in the manuscript, there was a lack of biological function research except for the gene expression profiles of the male and female flowers at different developmental stages. I thought simple bioinformatic analyses were not enough, and could not meet the requirement of the journal (Forests). Some wet labs about the MAD-box gene family were essential for the research. 

Reviewer 2 Report

The authors have conducted the systematic analysis of MADS-box family in Ginkgo biloba. The study identified the 26 MADS-box genes via genome-wide analysis of the genome. Authors have done some serios of bioinformatic analyses to support their current study. Further, the authors revealed that the results are essential to enhance understanding of the MADS-box gene family and provide a reference for studying the molecular mechanism of the development and differentiation of G. biloba flowers.

The manuscript written comprehensively and the theme of the study is presented appropriately.

Authors should check the formatting errors in throughout the manuscript. The gene name and style formatting are misleading.

Please describe the experimental materials in more detail, why choose ‘Jiafoshou’? Ginkgo is dioecious, how to get both female and male flowers?

Detailed chromosomal location prediction in TBtools, flow chart is needed for understanding. It should be given in response to reviewer’s letter.

Is there any specific reason or rationale for GbMADS protein and Gnetum, Oryza and Arabidopsis sequences for phylogenetic tree construction? I am just curious about that; the genus and genome of these species are different.

The methods section should contain a framework figure of listing all the analysis/steps done in this paper.

I suggest the authors give photos to illustrate the developmental state at the time of sampling (OS1/2/3 and M1/2/3).

The lines in Figure S1 are very illegible.

Further authors must explain why each item of methodology was done in the methods section. So, that it can attract the wide research groups.

I suggest authors add hypothesis of the study and future directions.

Round 2

Reviewer 2 Report

I think the revision of the article are appropriate and acceptable for publication.